# One Size Fits All—Venomics of the Iberian Adder (*Vipera seoanei*, Lataste 1878) Reveals Low Levels of Venom Variation across Its Distributional Range

**DOI:** 10.3390/toxins15060371

**Published:** 2023-06-01

**Authors:** Ignazio Avella, Maik Damm, Inês Freitas, Wolfgang Wüster, Nahla Lucchini, Óscar Zuazo, Roderich D. Süssmuth, Fernando Martínez-Freiría

**Affiliations:** 1CIBIO, Centro de Investigação em Biodiversidade e Recursos Genéticos, InBIO Laboratório Associado, Campus de Vairão, Universidade do Porto, 4485-661 Vairão, Portugal; inesfreitas@cibio.up.pt (I.F.); nahla.lucchini@cibio.up.pt (N.L.); 2Departamento de Biologia, Faculdade de Ciências, Universidade do Porto, 4099-002 Porto, Portugal; 3BIOPOLIS Program in Genomics, Biodiversity and Land Planning, CIBIO, Campus de Vairão, 4485-661 Vairão, Portugal; 4Institut für Chemie, Technische Universität Berlin, Straße des 17. Juni 124, 10623 Berlin, Germany; maik.damm@tu-berlin.de (M.D.);; 5Molecular Ecology and Evolution at Bangor, School of Natural Sciences, Bangor University, Bangor LL57 2UW, UK; w.wuster@bangor.ac.uk; 6Calle La Puebla 1, 26250 Santo Domingo de la Calzada, Spain

**Keywords:** snake venom, viper, bottom-up proteomics, regional variation, Iberian Peninsula

## Abstract

European vipers (genus *Vipera*) are medically important snakes displaying considerable venom variation, occurring at different levels in this group. The presence of intraspecific venom variation, however, remains understudied in several *Vipera* species. *Vipera seoanei* is a venomous snake endemic to the northern Iberian Peninsula and south-western France, presenting notable phenotypic variation and inhabiting several diverse habitats across its range. We analysed the venoms of 49 adult specimens of *V. seoanei* from 20 localities across the species’ Iberian distribution. We used a pool of all individual venoms to generate a *V. seoanei* venom reference proteome, produced SDS-PAGE profiles of all venom samples, and visualised patterns of variation using NMDS. By applying linear regression, we then assessed presence and nature of venom variation between localities, and investigated the effect of 14 predictors (biological, eco-geographic, genetic) on its occurrence. The venom comprised at least 12 different toxin families, of which five (i.e., PLA_2_, svSP, DI, snaclec, svMP) accounted for about 75% of the whole proteome. The comparative analyses of the SDS-PAGE venom profiles showed them to be remarkably similar across the sampled localities, suggesting low geographic variability. The regression analyses suggested significant effects of biological and habitat predictors on the little variation we detected across the analysed *V. seoanei* venoms. Other factors were also significantly associated with the presence/absence of individual bands in the SDS-PAGE profiles. The low levels of venom variability we detected within *V. seoanei* might be the result of a recent population expansion, or of processes other than directional positive selection.

## 1. Introduction

Snake venoms comprise a mixture of proteins, peptides, organic molecules (e.g., carbohydrates, lipids), and salts in an aqueous medium [1,2], and are able to disrupt the homeostatic processes of the envenomated organism [3,4,5]. To date, more than 60 protein families that compose snake venoms have been identified [6]. This complexity of components harbours the potential for extreme variation, which has been found to occur frequently and at all taxonomic levels (see [7,8,9]). Indeed, broad compositional differences have been found between venoms of snakes belonging to different families (e.g., Elapidae and Viperidae [6,10]), between different genera belonging to the same family (e.g., Australian elapids [11,12]; Old World vipers [13]), and between congeneric species (e.g., within the genera *Bothriechis* [14] and *Bothrops* [15]). Moreover, variation in venom composition and activity has also been found to occur at the intra-specific level, between juveniles and adults (e.g., [16,17,18]), between sexes (e.g., [19,20,21]), and between individuals from different geographic localities (e.g., [22,23,24]), where it may persist even in the face of extensive gene flow [25].

Extensive evidence suggests that the primary function of snake venoms is prey subjugation, and that the main driver behind the dynamism of its variation is adaptation to diet [26,27,28]. In this scenario, the occurrence of venom variation between conspecific snakes originating from different areas (i.e., geographic/regional variation) has generally been associated with differences in their feeding ecology (e.g., [28,29]. Indeed, specimens living in environmentally distinct areas are likely under different ecological pressures (e.g., different prey availability), which likely determines changes in their diet. Considering the critical adaptive value and fast evolutionary rates of snake venom [30,31], it stands to reason that these changes might drive the emergence of compositional and/or functional venom variation [8].

Multiple studies have then supported the adaptive value of geographic variation in the venom composition of different snake taxa in the face of prey type, availability, and/or susceptibility to venom (e.g., [32,33]), and a number of them found correlations between the occurrence of snake venom variation and different environments. For example, Sousa et al. (2017) found that *Bothrops atrox* venoms of specimens from different habitats within the Amazon presented significantly different activities [34]. Similarly, Zancolli et al. (2019) found a strong correlation between the dramatic venom variation occurring in *Crotalus scutulatus* [35,36] and environmental heterogeneity [25]. A correlation between venom phenotype and climatic variables in the same species was also found by Strickland et al. (2018) [37]. In these studies, the authors hypothesised that climatic/environmental variables could be determining changes in factors such as snake prey distribution and/or physiology, likely affecting the feeding ecology of the snake species, thus ultimately supporting adaptation to diet as a possible cause of the detected venom variation.

European vipers (genus *Vipera*) are among the snakes notable for extensive venom variation, including at the interspecific level (see [13]), as well as within some species. In particular, such variation corresponds to the occurrence of profound clinical differences between geographically close populations of species such as *Vipera aspis* and *Vipera berus* [38,39]. However, several *Vipera* species remain inadequately studied, especially in terms of the presence of intraspecific venom variation, which represents a significant knowledge gap given the medical importance of the genus [40,41,42].

One of the hitherto neglected species of *Vipera* is the Iberian adder, *Vipera seoanei*, Lataste 1879, a venomous snake belonging to the family Viperidae (subfamily Viperinae) and a member of the clade of vipers with Euro-Siberian affinity (i.e., *Pelias* [43]). Endemic to the northern Iberian Peninsula and south-western France, this species mainly inhabits areas with an Atlantic climate, typically occupying humid ecotones between meadows and forests and zones with abundant basal vegetation from sea level to about 1900 m of altitude (i.e., Cantabrian Mountains [44]). Sister species of the common adder *V. berus* (see [43]), *V. seoanei* displays low intraspecific genetic variability, likely as result of a late Pleistocene expansion from north-western Iberian refugia [45]. Despite its shallow genetic structure, *V. seoanei* shows considerable variation in biometric and pholidotic traits across its range [46,47]. The species is notable for high levels of polymorphism in body colouration, with five colour phenotypes currently recognised (i.e., *bilineata*, *cantabrica*, *classic*, *melanistic*, *uniform*). These appear to be geographically structured and not concordant with mitochondrial DNA haplotypes [45,48].

The diet of *V. seoanei* mainly comprises small mammals and, less frequently, reptiles, amphibians, arthropods, and birds [49,50,51]. Like many other species of the genus *Vipera* (e.g., *V. aspis* [52]; *Vipera latastei* [53]), *V. seoanei* exhibits ontogenetic shift in diet composition, with ectotherms (e.g., reptiles, amphibians) constituting more than 70% of the diet of juvenile vipers, whereas small mammals (i.e., shrews and rodents) account for roughly 90% of the diet of the adults [50]. Interestingly, significant correlations have been found between climatic and habitat conditions and differences in the frequency of consumption of the prey items *V. seoanei* mainly feeds on (i.e., amphibians, reptiles, and small mammals [50]). Such environment-related variation in diet composition, also observed in other *Vipera* species (e.g., *Vipera ammodytes* [54]; *V. latastei* [55]), suggests the presence of dietary differences across the species range.

The Iberian adder is recognised as medically important by the World Health Organization [42] and is considered one of the venomous snake species of major clinical relevance in Europe [40,41]. An early study [56] aiming at investigating the toxicity of *V. seoanei* venom across its Spanish distribution suggested the presence of a West–East gradient, with venoms from the western populations presenting higher lethal potencies than those collected from eastern populations (e.g., Galicia (W Spain): LD_50_ = 6.9–9.9 μg per 20 g mouse; Basque Country (E Spain): LD_50_ = 23.1–23.6 μg per 20 g mouse). A more recent study [57] reported toxicity values for *V. seoanei* of Portuguese origin (i.e., LD_50_ = 9.7 μg per 18–20 g mouse) comparable to those previously obtained by Detrai et al. (1990) [56] for individuals from Galicia and north of León, potentially supporting the higher toxicity of the westernmost populations. Interestingly, although *V. seoanei* venoms collected from different localities across the species’ Spanish distribution presented differences in LD_50_ values, the corresponding proteinograms appeared to be very similar, suggesting low levels of interpopulational divergence in venom compositions [56]. In spite of this intriguing scenario, and of the recognised medical relevance of *V. seoanei*, no other studies regarding this species’ venom have been conducted, and further information about it is currently unavailable.

In the present work, we (i) provide the first proteomic characterisation of the venom of *V. seoanei*; (ii) assess the level of venom variation across the viper’s range in the Iberian Peninsula; (iii) test for associations between biological, geographic, genetic, and eco-geographic factors and venom variation within this species across its ecologically and physiographically diverse range.

## 2. Results

### 2.1. Protein Composition of V. seoanei Venom

A total of 55 peaks were identified within the chromatographic profile of the *V. seoanei* venom pool (Figure 1A). The chromatogram shows several broad peaks with high abundance (e.g., peaks 6, 26, 32, 33, 36, 40, 54, and 55) and complex regions with smaller peaks, e.g., eluting between 35 and 50 min, as well as between 80 and 95 min. The subsequent reducing SDS-PAGE analysis revealed that the highest HPLC peaks are often dominated by few, highly abundant bands (see Figure 1B).

The major toxin families in the venom proteome included phospholipases of type A_2_ (PLA_2_) and snake venom serine proteases (svSP), followed by the slightly less abundant disintegrins (DI), snake venom C-type lectins and C-type lectin-related proteins (snaclec), and snake venom metalloproteinases (svMP). Other toxin families such as vascular endothelial growth factors (VEGF), cysteine-rich secretory proteins (CRISP), L-amino-acid oxidases (LAAO), Kunitz-type inhibitors (KUN), and venom nerve growth factors (NGF) were also detected with relative abundances of 6% or less. The peptidic part, made of inhibitors of svMPs (svMP-i), natriuretic peptides (NP), and other peptides composed about 11% of the venom proteome. Non-annotated proteins (NA) accounted for less than 0.2% of the venom proteome. For further details, see Figure 2 and Appendix A.

Among the PLA_2_s, we detected a variety of different proteoforms, some highly similar to the non-enzymatic PLA_2_ homologue S49 ammodytin L(2) [Q6A394], and two enzymatic active D49 forms, namely ammodytin I1 and I2 [Q910A1;P34180]. Therefore, *V. seoanei* venom includes basic, neutral, and acidic PLA_2_s. The main svSP we identified (6%, peak 44–45) shows high similarity with nikobin [E5AJX2] and an RVV-Vγ homolog [P18965] (4%, peak 43–45). Similarities were also found with other svSPs, such as SP-4/5/8 isoforms [A0A1I9KNR8; A0A1I9KNR5; A0A6B7FPJ0] of *Vipera ammodytes ammodytes*.

All svMP fragments were annotated using MS as members of the PIII-svMP subfamily, including ions from SDS bands with molecular weights of 30 kDa or less. This range of lower molecular svMP masses usually originates from mature PI or processed PII fragments but is uncommon for PIII. Sequence analysis showed that peaks 28–30 refer only to the DC domain of a PIII-svMP, which is strong evidence for PIIIe, since no PII could be assigned [58,59]. The DIs detected in the venom of *V. seoanei* were mostly composed of homologs of dimeric RGD-disintegrins, such as VA6 [P0C6A5], lebein-1α [P83253], and VB7B [P0C6A7]. Snaclecs were constantly detected in two- or four-band patterns at late retention times of 85–110 min. These toxins are known to form multimeric structures with one, two (αβ), or four (αβγδ) subunits in different complexities, such as (αβ)4 heterooctamers, and in combination with svMP P-IIIe [58,60]. We identified several homologs to different snaclec subunits isolated from *Vipera ammodytes ammodytes* and from the Eurasian viper genera *Daboia* and *Macrovipera* (see Appendix A).

The less abundant toxins include two classes of growth factors, namely NGF with two populations of 15 kDa and VEGF, almost exclusively identified as Vammin-1 homologs [APB93447]. A single band population of 25 kDa (peak 36–37) contains CRISP homologs of *V. berus* CRVP [B7FDI1]. The peptide part (11%) of the produced venom proteome is dominated by small tripeptidic svMP-is, as is the case for most peptidomes of Old World viper venoms [13]. Also included in the peptidic part are natriuretic peptides and fragments of bradykinin-potentiating and C-type natriuretic peptides (see Appendix A).

### 2.2. Assessing Geographic Venom Variation

#### 2.2.1. Analysis of the SDS-PAGE Profiles

In total, 49 SDS-PAGE whole venom profiles were produced from the individual *V. seoanei* venoms collected (Appendix A). By visually comparing the obtained profiles, we did not detect stark differences in terms of the presence/absence of bands, and instead noticed a considerable overall level of similarity between them. The least diverse profiles had 13 evident bands, whereas the most diverse had 18. Six bands were difficult to identify reliably. Nine bands were present in all venom profiles so were uninformative, and were thus excluded from further analyses.

#### 2.2.2. Band Analysis and NMDS

The final binary matrix (Appendix A) used to assess the diversity between the 49 SDS-PAGE whole venom profiles included 10 polymorphic (i.e., not appearing in every profile) bands. Their number varied between four (samples 19VS076, 19VS185, 21VS005, and 21VS006) and nine (samples 19VS450, 19VS451, and 21VS022) across the profiles (see Appendix A).

The three-dimensional NMDS analysis achieved an overall stress value of 0.08, indicating that the method could suitably represent the differences between the venom profiles. The NMDS ordination plot generated using the individual NMDS scores on the first two axes (i.e., NMDS1 and NMDS2; Figure 3) showed low dispersion of the samples across the defined space and a considerable level of overlap.

#### 2.2.3. Regression Analysis

Of the 14 predictors tested using the NMDS1 scores as the response variable, only four were significant in univariate linear regression analyses, namely SVL (*p* = 0.016), COLOUR (*p* = 0.028), POPULATION (*p* = 0.001), and FOREST (*p* = 0.035). In the univariate linear regression analyses testing the effect of each of the 14 predictors on the NMDS2 scores, only the three predictors SEX, COLOUR, and FOREST were significant (*p* = 0.022, *p* = 0.041, and *p* = 0.033, respectively). Details of the univariate linear regression models performed are reported in Table 1.

In the multiple regression model comprising individual NMDS1 scores as the response variable and SVL, COLOUR, and FOREST as predictors, no significant effects were detected. In the multiple regression model built including individual NMDS2 scores as the response variable and the three predictors SEX, COLOUR, and FOREST, only SEX had a significant effect (*p* = 0.045). The *p*-value relative to Moran’s I calculated for all variables included in the two multiple regression models was >0.1, indicating the absence of statistically significant spatial autocorrelation. Details of the multiple regression models used are reported in Table 2.

Following the same approach, we used single-predictor generalised linear models (GLMs) to test the effect of each of the 14 selected predictors on the presence/absence of each of the 10 polymorphic bands retrieved from the individual whole venom profiles. While no significant effects on the presence/absence of bands 2, 5, and 6 were detected, at least one predictor was significantly correlated with the presence/absence of the remaining seven polymorphic bands. The effect of only one predictor was significant for bands 1 (i.e., SVL, *p* = 0.022) and 3 (i.e., POPULATION, *p* = 0.021); thus, we could not perform multiple-predictor GLMs for these two bands. Similarly, we could not build a multiple-predictor GLM for band 7 because single-predictor GLMs used for this band found significance only for the predictors COLOUR (*p* = 0.002) and POPULATION (*p* = 0.026), which could not be included in the same model because of the high level of association between them (Cramér’s V = 0.899). Of all single predictor-GLMs used, some provided unreliable fit and low prediction power and, thus, were not considered in subsequent analyses. Additional details of the single-predictor binomial GLMs used are reported in Appendix A.

Multiple-predictor GLMs were built for bands 4, 8, 9, and 10 by including all the predictors that had a significant effect in the single-predictor GLMs for these four bands. Significant effects were found for the predictors SEX (*p* = 0.018) and AGRIC (*p* = 0.044) on the presence/absence of band 4, and for the predictor COLOUR (*p* < 0.001) on the presence/absence of band 10 (Table 3). Specifically, band 4 was significantly less detected in females than in males (Figure 4A), and the probability of detecting it appeared to be inversely related to the percentage of agricultural areas (Figure 4B). Concerning the significant effect of the predictor COLOUR on the presence/absence of band 10, the four *V. seoanei* colour phenotypes considered (i.e., *bilineata*, *cantabrica*, *classic*, *melanistic*) were related differently to the response variable (Figure 4C).

The effect of predictors SEX and GEN2 on the presence/absence of band 9 was also significant, but the multiple-predictor GLM for this band showed unreliable fit and low prediction power and was, therefore, not considered. None of the predictors included in the model with the presence/absence of band 8 as the response variable were significant.

The *p*-value relative to Moran’s I calculated for all variables included in the four multiple-predictor GLMs was >0.1, meaning that there was no statistically significant spatial autocorrelation in the data.

## 3. Discussion

### 3.1. Protein Composition of V. seoanei Venom

Through the application of bottom-up venomics, we were able to provide the first characterisation of the protein components present in the venom of *V. seoanei*. As shown by the high number of peaks and bands retrieved from the RP-HPLC and SDS-PAGE profiles produced for the venom pool (see Figure 1), the Iberian adder’s venom comprises at least 12 different toxin families. Five of these families, namely PLA_2_, svSP, DI, snaclec, and svMP, constitute about 75% of the full venom proteome (see Figure 2 and Appendix A). Similar compositional patterns have been described for the venoms of *V. seoanei*’s sister species *V. berus* [61] and the Iberian endemic *V. latastei* [16], in line with the general composition of Viperinae venoms, of which the four toxin families PLA_2_, svSP, snaclec, and svMP typically compose 60–90% [13].

The effects generally caused by the five major toxin families composing the venom of *V. seoanei* are concordant with the mainly haemorrhagic and cytotoxic symptoms typically reported for viper envenomation [62,63]. Specifically, DI, snaclec, and svSP can affect blood coagulation, fibrinolysis, angiogenesis, and platelet aggregation [64,65,66], and svMP (especially class PIII) are known to affect the coagulation cascade and platelet aggregation and to cause severe haemorrhage [67,68]. Phospholipases of the PLA_2_ type, the most abundant component of the analysed venom pool, constitute a very diverse toxin family [69] that can produce a plethora of different effects, including myotoxicity, cardiotoxicity, cytotoxicity, and coagulotoxicity [70]. Of the several different PLA_2_ proteoforms we detected in the *V. seoanei* venom, some of them appear to correspond to ammodytin L and ammodytin I, suggesting myotoxic and haemolytic effects (see [69]).

Among the less abundant toxins we found in *V. seoanei* venom, KUN and LAAO have been reported to interfere with platelet aggregation, fibrinolysis, and angiogenesis [71,72,73], with the latter also potentially being disrupted by CRISP and VEGF [74,75]. Finally, NGF can increase vascular permeability in the envenomated organism, thus aiding the spread of other toxins [76], while NP can decrease myocardial contractility and cause hypotension, rapidly leading to the loss of consciousness [77].

### 3.2. Low Levels of Geographic Variation

In order to assess the presence and extent of geographic venom variation within *V. seoanei*, we performed comparative analyses of the SDS-PAGE whole venom profiles obtained from the 49 individuals collected across the species’ distribution in northern Iberia. By visually comparing the profiles in question (Appendix A), we noticed the lack of marked differences among them. Additionally, in the NMDS ordination plot generated using the individual NMDS scores on the first two axes (i.e., NMDS1 and NMDS2), most of the points were clustered in an almost central position, several of them were overlapping, and we did not detect a defined pattern of geographic venom variation (see Figure 3). Although the Shapiro–Wilk test performed for NMDS1 and NMDS2 values showed that only the latter did not depart significantly from normal distribution (*W* = 0.951, *p* = 0.041 and *W* = 0.966, *p* = 0.168, respectively), the statistical models supported the classification of NMDS1 and NMDS2 values as normally distributed (41% of probability, against 20% of probability of following a beta distribution). Taken together, these results suggest the presence of low levels of variation within the adult *V. seoanei* venoms considered.

Being an ecologically critical functional trait, snake venom is under strong selective pressures, shaping its composition and activity to facilitate the snake’s survival [1,8]. The numerous reported cases of snake venom varying between different areas, likely in response to differences in ecological and environmental conditions (e.g., prey availability and susceptibility to venom), appear to provide consistent support for the adaptive value of snake venom variation (see [27,32]). Nonetheless, cases of geographic snake venom variation being almost undetectable at the intraspecific level are known. For example, Hofmann et al. (2018) and Rautsaw et al. (2019) did not detect a defined pattern of geographic variation in the venoms of *Crotalus cerastes* from Arizona and California, possibly due to stabilising selection favouring generalist venom phenotypes [78,79,80]. Similarly, Margres et al. (2015) found no significant variation in the expression of toxins and toxin genes across individuals of *Micrurus fulvius* from Florida, perhaps as result of relaxed selective constraints or a recent range expansion [81]. Therefore, it would be interesting to investigate if the low variation we observed among the SDS-PAGE *V. seoanei* venom profiles analysed could be a consequence of factors potentially preventing local venom adaptation, such as the recent population expansion from the north-west towards the eastern Iberian Peninsula suggested for this species [45], or considerable levels of gene flow (see [79]).

Additionally, in light of the arms race between snakes and their prey, typically consisting of prey evolving resistance to venom and snake venom evolving to bypass this resistance [82,83,84], it has been suggested that balancing selection might favour a diverse set of venom alleles over a single optimal venom genotype in order to prevent fixed venom alleles from becoming ineffective due to evolved prey resistance [36]. In this scenario, considering that adult Iberian adders are small mammal specialists [50], the low venom variation detected might align with balancing selection acting to allow the effective subjugation of their potentially coevolving preferred prey while also avoiding evolutionary dead ends.

### 3.3. Correlates of Venom Variation

In our regression models, we implemented the same predictors Espasandín et al. (2022) used to study the effect of eco-geographic variables on the trophic ecology of *V. seoanei* [50]. Considering the strong link between snake venom variation and diet, we aimed to test whether they could be at play in determining the occurrence and extent of venom variation within this species. Although we detected low overall levels of venom variation, the univariate linear regression analyses performed showed significant effects of body size (i.e., SVL), colour phenotype, sex, locality of origin of the vipers, and percentage of forested area on the NMDS1 and NMDS2 scores (see Table 1 for details). Nonetheless, in the multiple regression models, only the predictor SEX had a significant effect on the variation detected across the analysed venoms (see Table 2).

In gape-limited animals such as snakes [85], sexual dimorphism in body and/or head size can define the spectrum of prey items each sex can feed on [86]. While no sexual dimorphism concerning head size has been detected in *V. seoanei*, significant intersexual variation in body size (i.e., SVL) has been reported for this species [46]. Interestingly, significant differences in feeding frequencies have been found between male and female Iberian adders, with females feeding more often than males, and males reducing their feeding rates as they grow [50]. Although these considerations might suggest a potential influence of intersexual differences in feeding ecology on the low variation detected across the *V. seoanei* venoms analysed, we did not identify sex-specific venom protein bands. Additionally, despite the NMDS ordination plot indicating less divergence within female venoms than within male venoms (see Figure 3), the male and female venom profiles did not differ significantly in their complexity (i.e., number of bands; Mann–Whitney *U* = 284, *p* = 0.991). The significance of the sex and body size of the vipers could, therefore, be related to factors not included in our analyses (e.g., seasonality, reproductive stage), which might be unravelled by studies developed on a finer scale.

Considering the strong correlation between the two predictors POPULATION and COLOUR (see Section 5.5), and that the five colour phenotypes currently recognised for *V. seoanei* are geographically structured, the significance of these two predictors in our analyses might refer to different local selective regimes acting on the analysed venoms. For instance, individuals presenting the four colour phenotypes included in our analyses, i.e., *bilineata*, *cantabrica*, *classic*, and *melanistic*, are known to differ in body proportions, likely due to them being subjected to different ecological pressures (e.g., climatic conditions, predatory pressure) in the habitats where they occur [46]. Additionally, we suspect the significance of the predictor FOREST to be related to changes in the feeding ecology of *V. seoanei* associated with this habitat type (e.g., the consumption of amphibians by Iberian adders appears to increase in rainy and forested areas [50]). Considering the lack of marked genetic distinctness within *V. seoanei*, these results could hint that the little venom variation we detected might be related to ecological factors determining differences in local selective pressures acting on distinct colour phenotypes, and/or local changes in prey availability.

The single-predictor binomial generalised linear models (GLMs) used for the 10 polymorphic SDS-PAGE bands showed that the presence/absence of eight bands (i.e., 1, 2, 3, 4, 7, 8, 9, 10) was significantly correlated to at least one of the 14 predictors tested. Specifically, the models showed variation in band presence/absence in relation to body size (SVL), sex (SEX), colour phenotype (COLOUR), locality of origin (POPULATION), the PC2 of the SPCA of the interpolated genetic distances (GEN2), maximum temperature of the warmest month (BIO5), annual precipitation (BIO12), and the amount of cultivated fields (AGRIC) and forested area (FOREST) (see Appendix A). Multiple-predictor GLMs for bands 4, 8, 9, and 10 supported the significance of the predictors SEX and AGRIC for band 4 and of the predictor COLOUR for band 10, but did not provide significant or reliable results for bands 8 or 9 (see Table 3).

Based on the molecular weight of the corresponding bands in the SDS-PAGE profile of the venom pool (Figure 1), we suspect that bands 1, 2, 3, 4, and 10 of the whole venom SDS-PAGE profiles are likely to represent svMPs. Based on the same criterion, the content of the remaining five bands is more difficult to identify, since they likely include different toxin families (e.g., CRISP, LAAO, svSP). It is interesting to note that while the probability of occurrence of bands 2 and 10 increases with the amount of forested area (FOREST) and with the increase in the maximum temperature of the warmest month (BIO5), respectively, the probability of occurrence of band 4 is negatively correlated with the amount of cultivated fields (AGRIC; see Figure 4 and Appendix A). Snake venom metalloproteinases are thought to allow fast prey subjugation [27,67,68] and possibly aid prey digestion (controversial; see [87]). In a number of snake species undergoing an ontogenetic dietary shift from an ectotherm-based to an endotherm-based diet, smaller, younger individuals have been shown to produce more svMPs than larger, older specimens (e.g., *V. latastei* [16]; *Bothrops asper* [17]; *Crotalus viridis* [18]). While the number of ectotherms consumed by *V. seoanei* decreases as the viper grows, adult Iberian adders were found to consume more reptiles and amphibians in warm habitats and forests, respectively, and to feed on small mammals more frequently in agricultural areas [50]. In this scenario, the results of the single-predictor GLMs appear to at least partially support the importance of svMPs in subduing amphibians and reptiles. Indeed, the probability of occurrence of svMP-related bands 2 and 10 appears to be positively correlated with conditions that might favour an increase in the consumption of ectotherm prey (i.e., BIO5, FOREST). Conversely, the probability of detecting svMP-related band 4 decreases with the increase in conditions that have been linked to an increase in the consumption of endotherm prey by *V. seoanei* (i.e., AGRIC). Therefore, it could be interesting to investigate the toxins actually comprising these SDS-PAGE whole venom profile bands and test whether their presence provides any functional advantages in the subjugation of one prey type rather than the other (e.g., [88,89]).

## 4. Conclusions

This study provides the first proteomic characterisation of the venom of *Vipera seoanei*. Our results show that it presents the typical compositional pattern of Viperinae venoms, with the five toxin families PLA_2_, svSP, DI, snaclec, and svMP being predominant. Surprisingly, we found little variation among the SDS-PAGE profiles of the 49 considered venoms, which appeared instead to be remarkably similar. By performing regression analyses, we discovered significant effects of body size, colour phenotype, sex, locality of origin of the vipers, and percentage of forested area on the low levels of venom variation detected. In light of the highly dynamic scenario of snake venom variation, the information gathered here introduces new questions as to the selective drivers that underlie venom evolution in the Iberian adder. The development of genomic and functional studies could help us understand if the similarities we detected across the analysed venoms are the result of factors such as balancing selection and stabilising selection, or the by-product of a recent population expansion. Finally, proteomic analyses developed on a finer scale could likely unravel patterns of individual venom variation (e.g., sex-related, age-related, etc.) that remain undiscovered in *V. seoanei*.

## 5. Materials and Methods

### 5.1. Sampling

Between 2018 and 2021, a total of 49 individuals of *V. seoanei* were collected from 20 localities distributed across the species’ range in the Iberian Peninsula, with a maximum of five per locality (Figure 5; Appendix A). Venom samples were collected from each individual following the protocol reported by Avella et al. (2022) [16]. After venom extraction, tissue samples (i.e., buccal swabs) to be used for genetic analyses (see Section 5.4) were collected, and the sex and snout–vent length (SVL) of each snake were recorded. All vipers had an SVL > 325 mm, and were thus considered adults (see [46,90]).

Information about the colour phenotypes displayed by the collected vipers was also recorded, and each individual was assigned to one of the five categories reported by Martínez-Freiría et al. (2017) [48] (see Appendix A). Each viper was then released exactly where it had been captured. Venoms were lyophilised in a Scanvac (Coolsafe, Lynge, Denmark) freeze dryer and stored at −20 °C until being transported to the Süssmuth Laboratory of the Institut für Chemie, Technische Universität Berlin (Germany) for proteomic analysis.

Vipers, venoms, and tissue samples were collected with the permission of the Instituto da Conservação da Natureza e das Florestas, Portugal (ref. 537/2018, 362/2019, 295/2020 and 146/2021), Xunta de Galicia, Spain (ref. EB-017/2019, 018/2020 and 015/2021), Gobierno del Principado de Asturias, Spain (ref. 2019/003003, 2020/682020 and CO/09/017/2020), and Junta de Castilla y León, Spain (ref. EP/CyL/56/2018, 27/2019, 192/2020 and 54/2021).

### 5.2. Bottom-Up Venomics

#### 5.2.1. Venom Fractionation through RP-HPLC

Due to the low amount of venom we collected from each viper, we performed the proteomic analyses on a pool composed of equal amounts (i.e., 10 µL) of each of the 49 individual *V. seoanei* venoms. For reverse-phase chromatography (RP-HPLC), 1 mg of lyophilised venom from the pool was dissolved to a final concentration of 10 mg/mL in aqueous 5% (*v*/*v*) ACN with 1% (*v*/*v*) formic acid (HFo) and centrifuged for 5 min at 10,000× *g*. The supernatant was then fractionated using an HPLC Agilent 1200 (Agilent Technologies, Waldbronn, Germany) chromatography system equipped with a reversed-phase Supelco Discovery BIO wide Pore C18-3 (4.6 × 150 mm, 3 µm particle size) column. The following gradient with ultrapure water with 0.1% (*v*/*v*) HFo (solvent A) and ACN with 0.1% (*v*/*v*) HFo (solvent B) was used at 1 mL/min, with a linear solvent change given at min (B%): 0 (5%), 5 (5%), 100 (40%), 120 (70%), 130 (70%), and 5 min re-equilibration to 5% B. For monitoring the chromatography run, a diode array detector (DAD) was used at a λ = 214 nm detection wavelength. Samples were collected through time-based fractionation (1 fraction/min) and peak fractions were dried in a centrifugal vacuum evaporator.

#### 5.2.2. SDS-PAGE Profiling

The dried fractions were redissolved in 10 µL reducing 2× SDS sample buffer, heated for 10 min at 95 °C, and separated using 12% SDS-PAGE (SurePage Bis-Tris, Genscript, Piscataway, NJ, USA) run with MES buffer at 200 V for 21 min. A PageRuler Unstained Protein Ladder (Thermo Scientific, Waltham, MA, USA) was used as the protein standard. Gels were short-washed with water three times. Proteins were fixed three times for 10 min each with hot fixation buffer (aqueous, 40% (*v*/*v*) methanol, 10% (*v*/*v*) acetic acid), stained for 45 min in hot fast staining buffer (aqueous, 0.3% (*v*/*v*) HCl 37%, 100 mg/L Coomassie 250G) under constant mild shaking, and kept overnight at 4 °C in storage buffer (aqueous, 20% (*v*/*v*) methanol, 10% (*v*/*v*) acetic acid) for destaining. The produced gels were then scanned for documentation and quantification.

To produce profiles that could allow the assessment of similarities and differences among the 49 individual venoms, 20 µg of each lyophilised venom sample was loaded in 10 µL reducing 2× SDS sample buffer and subjected to SDS-PAGE profiling following the same protocol applied for the venom pool. The resulting gels were scanned for documentation, and the obtained digital images were used for statistical analysis. SDS-PAGE profiling was performed once for each venom sample.

#### 5.2.3. Tryptic Digestion

The bands of interest of the SDS-PAGE venom pool profile were cut, dried with 500 µL ACN, and stored at −20 °C until tryptic digestion. Disulphide bridges were reduced with 30 µL freshly prepared dithiothreitol DTT (100 mM in 100 mM ammonium hydrogen carbonate (ABC) per gel band) for 30 min at 56 °C and dried with 500 µL ACN for 10 min. Cysteines were alkylated with freshly prepared iodacetamid IAC (55 mM in 100 mM ABC) for 20 min at RT in the dark to protect the reduced thiols from oxidation and washed with 500 µL ACN for 2 min. Samples were dried with 500 µL ACN for 15 min, followed by 30 min incubation on ice with 20–30 µL freshly activated trypsin (13.3 ng/µL, 10% (*v*/*v*) ACN in 10 mM ABC; Thermo, Rockfeld, IL, USA). When necessary, additional volumes of trypsin were added. Samples were incubated for 90 min on ice, then 20 µL ABC buffer (10 mM) was added to all of them, and they were incubated overnight at 37 °C. Peptides were extracted with 100 µL elution buffer (aqueous, 30% (*v*/*v*) ACN MS grade, 5% (*v*/*v*) HFo) pre-warmed at 37 °C for 30 min. The supernatant was transferred into a separate microtube and vacuum-dried. Following a second HPLC purification of 1 mg crude venom, smaller dried fractions were submitted to LC-MS for direct peptide detection, without any SDS-PAGE separation or tryptic digestion.

#### 5.2.4. Mass Spectrometry

For the mass spectrometry (MS) analysis, the excised SDS-PAGE bands of interest were re-dissolved in 30 µL aqueous 3% (*v*/*v*) ACN with 1% (*v*/*v*) HFo, and 20 µL of each was injected into an Orbitrap XL mass spectrometer (Thermo, Bremen, Germany) via an Agilent 1260 HPLC system (Agilent Technologies, Waldbronn, Germany) using a reversed-phase Grace Vydac 218MS C18 (2.1 × 150 mm; particle size, 5 μm) column. The following gradient with ultrapure water with 0.1% (*v*/*v*) HFo (solvent A) and ACN with 0.1% (*v*/*v*) HFo (solvent B) was used at 0.3 mL/min, with a linear buffer change given at min (B%): 0 (5%), 1 (5%), 11 (40%), 12 (99%), 13 (99%), and 2 min re-equilibration to 5% B. The parameters in the ESI positive modus were as follows: 270 °C capillary temperature, 45 L/min sheath gas, 10 L/min auxiliary gas, 4.0 kV source voltage, 100.0 µA source current, 20 V capillary voltage, 130 V tube lens. FTMS measurements were performed with 1 μ scans and 1000 ms maximal fill time. AGC targets were set to 10^6^ for full scans and to 3 × 10^5^ for MS2 scans. MS2 scans were performed with a mass resolution (R) of 60,000 (at *m*/*z* 400) for *m*/*z* 250–2000. MS2 spectra were obtained in data-dependent acquisition (DDA) mode as top2 with 35 V normalized CID energy, and with 500 as the minimal signal required with an isolation width of 3.0. The default charge state was set to z = 2, and the activation time to 30 ms. Unassigned charge states and charge state 1 were rejected.

LC-MS/MS data RAW files were converted into the MASCOT generic file (MGF) format using MSConvert (Version 3.0.22187) with peak picking (vendor msLevel = 1−) [91]. For an automated database comparison, files were analysed using pFind Studio [92], with pFind (Version 3.1.5) and the integrated pBuild. The parameters used were as follows: MS Data (format: MGF; MS instrument: CID-FTMS); identification with Database search (enzyme: Trypsin KR_C, full specific up to 3 missed cleavages; precursor tolerance +20 ppm; fragment tolerance +20 ppm); open search setup with fixed carbamidomethyl [C] and Result Filter (show spectra with FDR ≤ 1%, peptide mass 500–10,000 Da, peptide length 5–100, and show proteins with number of peptides > 1 and FDR ≤ 1%).

The used databases included UniProt ‘Serpentes’ (ID 8750, reviewed, canonical and isoform, 2674 entries, last accessed on 10 February 2022; available at: https://www.uniprot.org/) and the common Repository of Adventitious Proteins (215 entries, last accessed on 10 February 2022; available at: https://www.thegpm.org/crap/index.html). The results were batch-exported as PSM score of all peptides identified with pBuild and manually cleared from decoy entries, contaminations, and artifacts. Finally, a list of unique peptide sequences per sample with the best final score was generated. For a second confirmation of identified sequences, all unique ones were analysed using BLAST search [93], with blastp against the non-redundant protein sequences (nr) of the Uniprot database “Serpentes” (taxid: 8570). In case of non-automatically annotated bands, files were checked manually using Thermo Xcalibur Qual Browser (version 2.2 SP1.4), *de novo* annotated, and/or compared on MS1 and MS2 levels with other bands to confirm band and peptide identities. Deconvolution of isotopically resolved spectra was carried out by using the XTRACT algorithm of Thermo Xcalibur.

#### 5.2.5. Relative Quantification of the Venom Pool Proteome

The quantification protocol is adapted from the three-step hierarchical venom proteome quantification protocol developed at the Evolutionary and Translational Venomics Laboratory of the Institute of Biomedicine of Valencia [94,95,96], based on a combination of the separation and quantification of HPLC peaks, SDS-PAGE bands, and ion intensities. Briefly, it defines the normalised toxin abundances within a single SDS-PAGE band, with the normalised values of the RP-HPLC peak integral measured at 214 nm, the gel band intensity, and, if necessary, the MS ion intensity of the most abundant peptides identified.

The gel band abundance measurement was performed through densitometry on the scanned gel images. Non-highly compressed PNG images of the SDS-PAGE gels were processed using the software Fiji [97]. Colour depth was changed to 8 bit grayscale, and the area and integrated density of each SDS-PAGE band were measured. The integrated relative densities were calculated using a representative background region as reference and normalised considering the area of the corresponding chromatographic peaks. In case of multiple toxin families identified within a single band, normalised toxin abundances were estimated based on the difference between the sum of the relative ion intensities of the three most abundant peptide ions of a toxin family and the sum of the three most abundant peptide ions of any other comigrated toxin family. Mass spectrometry proteomics data have been deposited with the ProteomeXchange Consortium10 via the MassIVE partner repository under the project name “Snake venomics of *Vipera seoanei*” with the dataset identifier “MSV000091535”.

### 5.3. Non-Metric Multidimensional Scaling

To assess the presence and extent of variation among the 49 *V. seoanei* venoms collected, we applied the individual-based approach reported by Zancolli et al. (2017) [98]. Thus, we generated a presence–absence matrix of the bands present in the individual SDS-PAGE venom profiles, excluding bands with frequency = 1 (i.e., not informative) or difficult to identify. We preferred analysing SDS-PAGE profiles rather than RP-HPLC profiles because in the latter, reliable peak identification can often be difficult due to venom complexity and differences in protein elution times. Indeed, during different RP-HPLC runs, the same proteins can present different elution times for a number of reasons (e.g., fluctuations in room temperature), further complicating the analysis of the chromatograms [98].

The final binary matrix (Appendix A) was used to analyse patterns of venom variation using non-metric multidimensional scaling (NMDS) based on pairwise Bray–Curtis similarity distances among profiles (see [99]). In order to keep the stress value below the 0.2 threshold (i.e., poor fit [100]), we opted for a three-dimensional NMDS analysis. We then used the individual NMDS scores on the first two axes (i.e., NMDS1 and NMDS2) to produce an ordination plot representing the dissimilarity between the individual SDS-PAGE profiles of the *V. seoanei* venoms analysed.

### 5.4. Predictors

To investigate which factors could potentially influence the occurrence of variation in the venoms of *V. seoanei*, we considered a total of 14 predictors: (i) three referring to biological traits of the vipers; (ii) one corresponding to the locality of origin of each viper (POPULATION); (iii) eight describing the climatic and habitat conditions of the geographic position where each viper was collected; and (iv) two corresponding to the first two principal components (GEN1 and GEN2) of a spatial principal component analysis (SPCA) performed for the interpolated mtDNA genetic distances estimated within *V. seoanei*.

The predictors related to the biological traits of the vipers were snout–vent length (SVL, in mm), sex (SEX), and colour phenotype (COLOUR) of each individual. The *uniform* colour phenotype, displayed by only one of the sampled individuals (i.e., 20VS165), was excluded from analyses involving the predictor COLOUR in order to avoid model performance hindering due to small sample size. The predictors related to the bioclimatic and habitat conditions of the geographic position where each viper was collected comprised four bioclimatic variables (i.e., annual mean temperature, BIO1; maximum temperature of the warmest month, BIO5; annual precipitation, BIO12; precipitation of driest month, BIO14) and the percentage of ground cover for four habitat types (i.e., cultivated fields, AGRIC; forest, FOREST; moors, MOOR; pastures and grasslands, PASTURE). These eight predictors, used in previous works on the ecology of *V. seoanei* and other *Vipera* species (e.g., [101,102]), have been found to influence the distribution and abundance of *V. seoanei* prey such as amphibians, reptiles, and small mammals (e.g., [103,104]), as well as their frequency in the species’ diet [50]. Values for bioclimatic predictors were extracted from raster layers at a resolution of 30 arc seconds (~1 km) from WorldClim version 2.1 ([105]; available at www.worldclim.org, accessed on 25 November 2022). Values for habitat types were extracted from Corine Land Cover version 2020_20u1 (available at https://land.copernicus.eu/pan-european/corine-land-cover/clc2018, accessed on 26 October 2022) after grouping land cover categories describing similar structural habitat types and upscaling the resulting rasters at a 2 km^2^ resolution (see [50]).

In order to obtain information on the genetic structure within *V. seoanei*, to be used as a predictor of venom variation in regression analyses, we estimated mtDNA genetic distances within this species. While *V. seoanei* sequences are available from previous works (see [45]), we aimed to obtain a better geographic coverage of the genetic diversity existing across the species’ range and, thus, decided to also consider genetic data obtained from the individuals we collected. To this end, total genomic DNA was extracted from buccal swabs from 20 vipers (1 specimen from each sampled locality; see Appendix A), using a standard saline method. Two mitochondrial (mtDNA) gene fragments, NADH dehydrogenase subunit 4 (ND4; 630 bp) and cytochrome b (cyt b; 554 bp), were amplified through polymerase chain reaction (PCR) using primers VSnd4-F/VSnd4-R [45] and CB1/CB2 [106], respectively. Laboratory procedures for DNA amplification and sequencing followed the protocols used in [45]. In order to produce a geographically comprehensive genetic distance matrix, we considered the sequences of 65 specimens, namely 20 retrieved from this study, 44 from GenBank, and one from a specimen included exclusively in the genetic analyses to enhance geographic coverage (specimen code 20VS008). Sequences were manually aligned and edited using Geneious v 4.8.5 [107]. Uncorrected p-distances between populations were estimated for the resulting alignment using MEGA X [108]. For information about the 65 specimens used to produce the genetic distance matrix and GenBank sequence accession numbers, see Appendix A.

The genetic distances calculated in this way were then spatially interpolated using the kriging method and summarised through spatial principal component analysis (SPCA). The first two of the resulting principal components, hereafter named GEN1 and GEN2, explained 54% and 40% of the variance, respectively (see Appendix A). GEN1 and GEN2 were used as predictors in the subsequent regression analyses, as a proxy of the genetic structure of *V. seoanei*. Values of GEN1 and GEN2 were obtained in ArcGIS version 10.5 by extracting information on the geographic position of each specimen from the original SPCA raster. Interpolations and SPCA were performed in ArcGIS version 10.5 [109], following the same procedure applied in [45]. The genetic distance matrix was calculated using the *ape* package [110] in R, version 4.2.2 [111].

### 5.5. Regression Analysis

We applied linear regression models to investigate the relationship between the predictors considered and the individual scores on the first two axes of the NMDS analysis performed on the polymorphic bands of the SDS-PAGE venom profiles. To this end, using NMDS1 and NMDS2 as response variables, we performed univariate linear regression models for each of the 14 predictors. In order to separate the effects of the different predictors tested, we then performed multiple regression models built using NMDS1 and NMDS2 as response variables and including all the predictors that were shown to be significant in the univariate linear regression models.

We then tested whether the presence or absence of the polymorphic bands identified in the venom SDS-PAGE profiles was significantly correlated with the predictors considered. Thus, we used single-predictor binomial generalised linear models (GLMs) for each predictor, considering the presence/absence of each band as a binomial dependent variable. Following the same approach described above, we then built multiple-predictor GLMs including all the predictors that were shown to be significant in the single-predictor GLMs.

In all models, all continuous predictors were scaled (i.e., mean = 0; SD = 1). The presence of spatial autocorrelation for all variables included in the models with more than one predictor was tested by calculating Moran’s I [112]. Correlation between the predictors included in the multiple regression models was generally low (<50%), but the predictors POPULATION and COLOUR showed a strong association (Cramér’s V = 0.899; see [113]). Since the number of levels of POPULATION (five times greater than COLOUR’s; i.e., 20 vs. 4) was more likely to hinder model performance (see [114]), it was excluded from multiple regression models and multiple-predictor GLMs including the predictor COLOUR.

We used the packages *vegan* [115] to perform the NMDS analysis, *car* to perform the regression models [116], and *ggeffects* [117] to plot model predictions. All analyses were performed in R, version 4.2.2 [108].

## Figures and Tables

**Figure 1 toxins-15-00371-f001:**
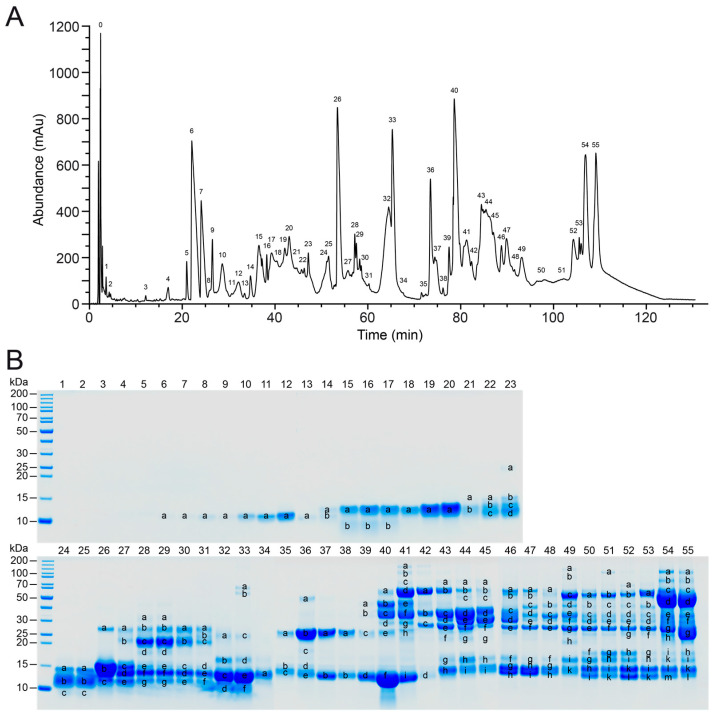
Fractionation of the *V. seoanei* venom pool. The figure shows RP-HPLC profile (**A**), with peak 0 corresponding to the injection peak, and SDS-PAGE Coomassie-stained profile (**B**) of the venom pool under reducing conditions. PAGE line nomenclature is based on RP-HPLC fractions. Labelled bands were cut, subjected to tryptic digestion, and analysed with LC-MS.

**Figure 2 toxins-15-00371-f002:**
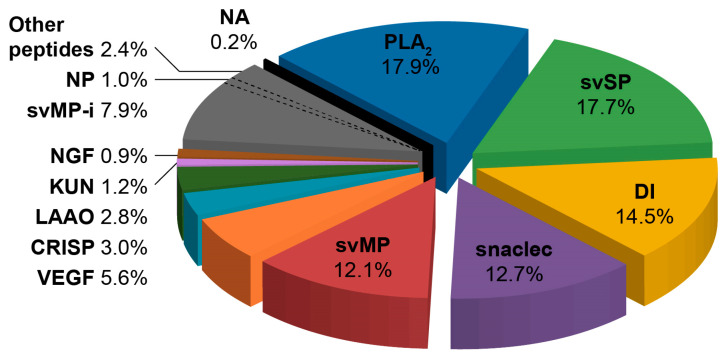
Reference composition of *Vipera seoanei* venom. The pie chart displays the relative abundances of the toxin families found in the proteome of the *V. seoanei* venom pool. PLA_2_, phospholipases A_2_; svSP, snake venom serine proteases; DI, disintegrins; snaclec, snake venom C-type lectins and C-type lectin-related proteins; svMP, snake venom metalloproteinases; VEGF, vascular endothelial growth factors; CRISP, cysteine-rich secretory proteins; LAAO, L-amino-acid oxidases; KUN, Kunitz-type inhibitors; NGF, venom nerve growth factors; svMP-i, svMP inhibitors; NP, natriuretic peptides; NA, non-annotated proteins. The sum of the percentages does not match 100% because of rounding. Detailed percentages are reported in Appendix A.

**Figure 3 toxins-15-00371-f003:**
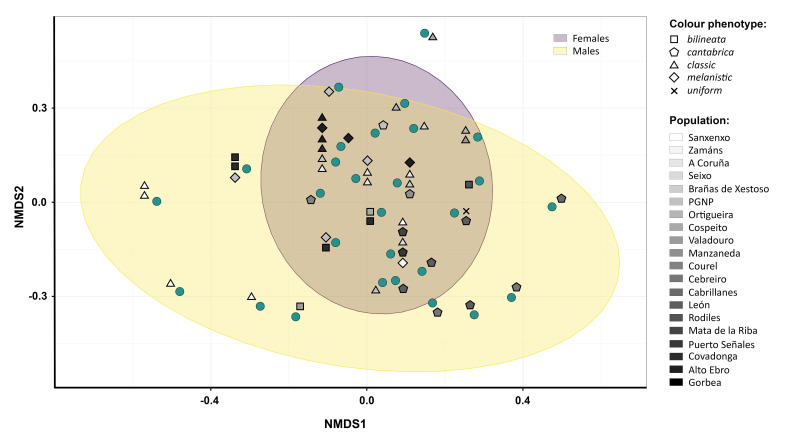
Non-metric multidimensional scaling (NMDS) ordination plot of the 49 *V. seoanei* venom SDS-PAGE profiles. Teal circles represent the venom profiles. Notice that some circles correspond to overlapping profiles (up to six).

**Figure 4 toxins-15-00371-f004:**
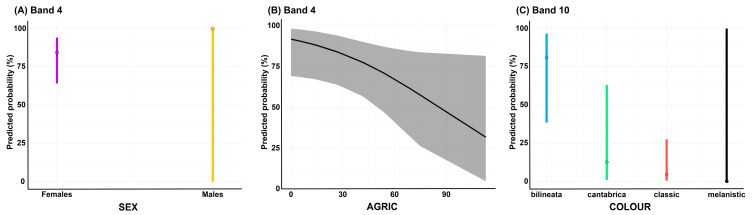
Multiple-predictor GLM predictions of occurrence of bands 4 and 10 in individual SDS-PAGE venom profiles. The panels display the predicted probability of occurrence of band 4 in relation to the sex of the viper (**A**) and amount of cultivated fields (**B**), and of band 10 in relation to four different colour phenotypes displayed by *V. seoanei* (**C**). The high standard deviations are attributable to model limitations owing to small sample size.

**Figure 5 toxins-15-00371-f005:**
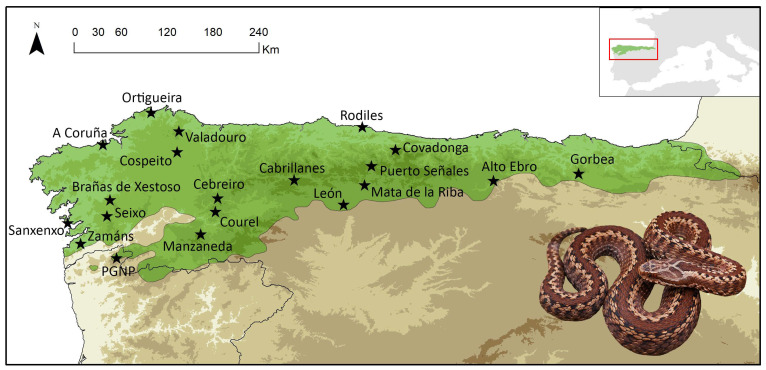
Map of the 20 localities sampled across the Iberian distribution of *V. seoanei*. Black stars indicate the sampled localities. The area in green represents the species’ full distributional range. Additional details about the sampling are reported in Appendix A.

**Table 1 toxins-15-00371-t001:** Results of the univariate linear regression models using the individual NMDS1 and NMDS2 scores. For each predictor tested, sum of squares (Sum Sq), degrees of freedom (Df), F value (*F*), and associated *p*-value (*p*) are reported. Significant predictors and corresponding *p*-values are in bold.

	NMDS1				NMDS2			
Predictor	Sum Sq	Df	*F*	*p*	Sum Sq	Df	*F*	*p*
**SVL**	0.263	1	6.351	**0.016**	<0.001	1	0.013	0.911
**SEX**	0.081	1	1.794	0.187	0.205	1	5.626	**0.022**
**COLOUR**	0.397	3	3.312	**0.028**	0.324	3	2.983	**0.041**
**POPULATION**	1.538	19	3.508	**0.001**	1.032	19	1.779	0.079
GEN1	0.029	1	0.635	0.429	<0.001	1	<0.001	0.997
GEN2	0.015	1	0.322	0.573	0.016	1	0.403	0.529
BIO1	0.061	1	1.334	0.254	0.001	1	0.031	0.861
BIO5	0.145	1	3.314	0.075	0.123	1	3.379	0.072
BIO12	0.133	1	3.019	0.089	<0.001	1	0.009	0.925
BIO14	0.002	1	0.048	0.823	0.015	1	0.363	0.549
AGRIC	0.016	1	0.345	0.559	0.102	1	2.646	0.111
**FOREST**	0.201	1	4.702	**0.035**	0.178	1	4.793	**0.033**
MOOR	0.002	1	0.051	0.823	0.004	1	0.107	0.746
PASTURE	<0.001	1	<0.001	0.981	<0.001	1	0.011	0.912

**Table 2 toxins-15-00371-t002:** Results of the multiple regression models. For each predictor included in the model, sum of squares (Sum Sq), degrees of freedom (Df), F value (*F*), and associated *p*-values (*p*) are reported. Residual sum of squares (Res. Sum Sq) and residual degrees of freedom (Res. Df) of the full models are also presented. Significant predictors and corresponding *p*-values are in bold.

Response	Predictor	Sum Sq	Df	*F*	*p*	Res. Sum Sq	Res. Df
NMDS1	SVL	0.084	1	2.142	0.151	1.641	42
	COLOUR	0.208	3	1.773	0.167		
	FOREST	0.021	1	0.536	0.468		
NMDS2	**SEX**	0.136	1	4.255	**0.045**	1.347	42
	COLOUR	0.223	3	2.317	0.089		
	FOREST	0.084	1	2.611	0.114		

**Table 3 toxins-15-00371-t003:** Results of the multiple predictor binomial GLMs. The models investigate the probability of occurrence of bands 4, 8, 9, and 10 in an SDS-PAGE venom profile. Likelihood ratio chi-square (LR *χ*^2^), degrees of freedom (Df), and associated *p*-values (*p*) are reported. Significant predictors and corresponding *p*-values are in bold. The asterisks (*) indicate predictors that, although significant, were not considered because the corresponding models did not converge.

Response	Predictors	LR *χ*^2^	Df	*p*
Band 4	**SEX**	5.544	1	**0.018**
	**AGRIC**	4.068	1	**0.044**
Band 8	SVL	0.529	1	0.467
	COLOUR	3.638	3	0.303
	BIO5	0.059	1	0.807
	BIO12	0.000	1	0.996
Band 9	**SEX ***	20.679	1	**<0.001**
	**GEN2 ***	16.784	1	**<0.001**
Band 10	**COLOUR**	16.612	3	**<0.001**
	BIO5	2.387	1	0.122
	FOREST	3.765	1	0.052

## Data Availability

Mass spectrometry proteomics data have been deposited with the ProteomeXchange Con-sortium10 via the MassIVE partner repository under the project name “Snake venomics of *Vipera seoanei*” with the dataset identifier “MSV000091535”. Additional data presented in this study are available in the Appendix A.

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
