# Peer review of "One Size Fits All—Venomics of the Iberian Adder (Vipera seoanei, Lataste 1878) Reveals Low Levels of Venom Variation across Its Distributional Range"

_toxins, 2023, doi:10.3390/toxins15060371_

Round 1

Reviewer 1 Report

Snake venom is a sophisticated weapon, composed mainly of proteins and peptides, used by snakes for prey immobilisation and defence. The composition of the venom varies greatly between snake families and also between genera and species of the same snake family. In addition, fine-tuned variations in venom composition have been observed at the species level, depending for example on the age and sex of the snake, the type of prey, the geographical location and the season.

In the present study, the authors report on the proteomic study of the venom of the Iberian adder (Vipera seoanei) and on the variations in the composition of the venom within a species. The latter is based on a comparative analysis of the SDS-PAGE profiles of venoms of wild adult snakes from different geographical regions. Using the standard bottom-up venomics approach (HPLC separation of the whole venom followed by SDS-PAGE analysis of fractions and gel-based MS identification of proteins), 12 protein and peptide families were identified in the venom of V. seoanei, of which the most abundant are PLA2s, SPs, disintegrins, snaclecs and MPs and their inhibitors. Thus, it shows great similarity to the proteomes of other venomous snakes of the family Viperidae, subfamily Viperinae. The comparison of the 49 SDS-PAGE venom profiles showed a very high similarity between the venoms studied, but also some differences, i.e. ten different protein bands appeared only in some profiles. The effect of biological, geographical, genetic and ecogeographical factors (a total of 14 selected predictors) on the presence/absence of each of the 10 polymorphic bands was tested by regression analysis. A correlation was found between body size, colour phenotype, sex and percentage of forested area with observed venom variation.

Before accepting the manuscript, I would like to ask the authors to answer the following questions.

General comments:

I propose to write 'snaclec' instead of 'CTL' as suggested by the Science and Standards Committee (Clemetson et al., 2008):

“To avoid confusion with classic C-type lectins and because names such as C-type lectin-like or –related proteins are nearly always abbreviated to CTL or CLP and do not convey any information about the heterodimeric structure, loop-swapping nor about higher order multimerization, by analogy with other classes such as siglecs (12), we propose to call this group snaclecs (Snake venom C-type lectins).”

I also suggest using the term ‘serine proteases’ instead of ‘serine proteinases’.

When authors cite the work of other researchers, in cases like this, I would prefer to give the name of the first author followed by the reference number.:

Line 62: change “For example, [34] found that …” to “For example, Sousa et al. (2017) found that … [34]”

or simply avoid that by saying “For example, Bothrops atrox venoms from specimens … [34].” Because it is not the reference that found or did something, but its authors.

This manuscript really lacks the 'the cherry on the cake', namely the identification of the proteins in the polymorphic SDS-PAGE bands. These should have been cut out and the proteins identified by MS. Generally, two-dimensional electrophoresis is a more suitable method for comparing proteomes than one-dimensional (SDS-PAGE). It is not only the protein family that matters, but also the occurrence of different proteoforms of a family, which often do not differ significantly in their molecular mass. When using sensitive staining protocols, the 2DE can be performed with only a few micrograms of snake venom, depending on the complexity.

Other comments:

Lines 131-132: If by 'dominant HPLC peaks' the authors mean the highest peaks on the chromatogram in Fig. 1B, e.g. 26, 30, 40, 54, 55, then it is not true that they consist predominantly of a single protein band.

Lines 156-158: The ammodytin L(2) isoform [Q6A394] is the S49 enzymatically inactive PLA2 homologue and the sentence should be amended accordingly.

Line 161: The ID numbers of the SPs of V. a. ammodytes are missing.

Lines 162-164: This sentence needs to be written more clearly.

Lines 191-193: The polymorphic bands need to be indicated in Fig. S1.

Line 293: Amend the sentence as follows:

… svMPs (especially class PIII) are known to affect the coagulation cascade and platelet aggregation, …

Line 353: Change ‘population of origin' to ‘locality of origin'.

Line 474: Change ‘buffer’ to ‘solvent’.

Reviewer 2 Report

The manuscript analyzes the individual venom of Vipera seoanei from the Northern Iberian Peninsula region. The proteomic was performed using a pool of 49 individual venoms collected. A pool of all individual venoms was characterized, and its composition was shown. SDS Page crude venom from individual snakes was analyzed. NMDS was applied to determine patterns of variation and correlation with other factors that could be related to venom profile. 

The first concern is why the authors chose to look at the SDS profile instead of the chromatograms of individual venoms. This has to be explicit in the article. Looking at chromatograms would facilitate the identification of the molecules that could correlate with other factors. For instance, which is the molecule that correlates with color?

The manuscript is well-presented and written, but I suggest that Figure S1 be included in Figure 3. This figure is essential for understanding the whole analysis of the band data. 

I am not an expert in these correlation analyses, but they are well presented, and data is solid, and the interpretation is satisfactory. It is a pity that the proteins related to color, sex, SVL, population, etc., could not be identified. Nevertheless, this analysis that, to my knowledge, was not yet used for snake venom can be of interest to many researchers in the area. 

Methods 

Line 467 Please specify the amount of each venom that composes the pool, was it the same amount for each individual venom, or were these amounts variable? 

Minor points

Specify in the legend of Figure 1B that it is a reducing SDS-PAGE-

The legend of Figure 3 appears twice. Lines 201 and 205.

Please introduce how many times the SDS Page of individual venom was performed. 
